# High-Resolution Sea Surface Temperatures Derived from Landsat 8: A Study of Submesoscale Frontal Structures on the Pacific Shelf off the Hokkaido Coast, Japan

**Hiroshi Kuroda [1],\* and Yuko Toya [2]**

1 Fisheries Resources Institute (Kushiro), Japan Fisheries Research and Education Agency, Kushiro 0850802, Japan
2 Fisheries Resources Institute (Sapporo), Japan Fisheries Research and Education Agency, Sapporo 0620922, Japan; toya_yuko82@fra.go.jp
\* Correspondence: kurocan@affrc.go.jp; Tel.: +81-154-92-1723

**Abstract:** Coastal and offshore waters are generally separated by a barrier or "ocean front" on the continental shelf. A basic question arises as to what the representative spatial scale across the front may be. To answer this question, we simply corrected skin sea surface temperatures (SSTs) estimated from Landsat 8 imagery with a resolution of 100 m using skin SSTs estimated from geostationary meteorological satellite Himawari 8 with a resolution of 2 km. We analyzed snapshot images of skin SSTs on 13 October 2016, when we performed a simultaneous ship survey. We focused in particular on submesoscale thermal fronts on the Pacific shelf off the southeastern coast of Hokkaido, Japan. The overall spatial distribution of skin SSTs was consistent between Landsat 8 and Himawari 8; however, the spatial distribution of horizontal gradients of skin SSTs differed greatly between the two datasets. Some parts of strong fronts on the order of 1 °C km$^{-1}$ were underestimated with Himawari 8, mainly because of low resolution, whereas weak fronts on the order of 0.1 °C km$^{-1}$ were obscured in the Landsat 8 imagery because the signal-to-noise ratios were low. The widths of the strong fronts were estimated to be 114–461 m via Landsat 8 imagery and 539–1050 m via in situ ship survey. The difference was probably attributable to the difference in measurement depth of the SST, i.e., about 10-μm skin layer by satellite and a few dozen centimeters below the sea surface by the in situ survey. Our results indicated that an ocean model with a grid size of no more than ≤100–200 m is essential for realistic simulation of the frontal structure on the shelf.

**Keywords:** sea surface temperature (SST); Landsat 8; Himawari 8; thermal front; submesoscale; continental shelf

## 1. Introduction

Sea surface temperature (SST) is one of the most essential variables in the study of oceanic, atmospheric, and marine ecosystem processes. In 1981, global measurements of high-quality SSTs began using an innovative multi-spectral sensor referred to as the Advanced Very High Resolution Radiometer (AVHRR) on board the National Oceanic and Atmospheric Administration (NOAA) polar-orbiting operational environmental satellites [1,2]. The spatial resolution was 1.1 km, and the sampling frequency was twice per day. Since the 1980s, a number of satellites with advanced sensors (e.g., Aqua, Terra, TRMM, MetOp, and Suomi NPP) have accurately monitored global SSTs. This monitoring has been essential for operational nowcasts/forecasts of ocean [3] and weather conditions [4–7].

A Japanese new generation geostationary meteorological satellite, Himawari 8, with state-of-the-art sensors "Advanced Himawari Imagers" began operational service in July 2015 [8,9]. The service phase was ahead of other third-generation geostationary meteorological satellites such as the GOES-16 (December 2017) and the Meteosat Third Generation (scheduled for 2021). The Advanced Himawari Imager has 16 observation bands. The spatial resolution is 2 km for the infrared band. The imager has capabilities comparable to those of the Advanced Baseline Imager on board GOES-16, which is run under a collaborative program between NOAA and the National Aeronautics and Space Administration (NASA) [10,11]. The greatest advance offered by Himawari 8 has been the high frequency of the Full Disk observations, i.e., every 10 min. High-frequency measurements have several important advantages compared with low-frequency samplings by non-geostationary satellites: high-frequency measurements make it possible to distinguish biases of skin SSTs between nighttime and daytime observations; they reduce cloud noise from skin SST; and they facilitate comparisons with data from other satellites with a temporal error of less than 10 min. Consequently, the skin SSTs from Himawari 8 are informative enough to enhance understanding of variations in the ocean [12–14] and enable appropriate initialization of an ocean forecast system [15].

This study made use of the skin SSTs from Himawari 8 and focused on ocean fronts, particularly thermal fronts on the continental shelf. In general, an ocean front is a relatively narrow transitional zone with enhanced horizontal gradients of physical, chemical, and biological properties (e.g., temperature, salinity, nutrient concentrations, and plankton distributions) that separates broader areas that differ in terms of vertical structure and stratification [16,17]. Ocean fronts can thus be regarded as barriers between water masses or ecosystems [18]. Many studies that have detected ocean fronts and examined their properties have used satellite-derived skin SSTs with a horizontal resolution of 1–25 km (e.g., [19–26]). The biggest advantage of using satellite data is that satellite sensors simultaneously measure much or even all of the horizontal structure of ocean fronts. This ability contrasts with the difficulties of making in situ measurements from ships. Shipboard measurements are constrained by the number and speed of available ships. Moreover, satellite-based frontal positions can be easily overlaid onto biological distributions. Sea mammals, birds, and some pelagic fish frequently aggregate around ocean fronts [27–30].

Since the late 2000s, oceanography has been revolutionized by the recognition of small-scale, short-term, so-called submesoscale processes in the ocean [31–34]. Generally, submesoscale variations in the ocean have timescales of days and spatial scales of 0.1–10 km. In this study, such time- and spatial scales are designated as $O(1)$ day and $O(0.1–10)$ km, respectively, where $O$ means "on the order of". These scales have been inferred mostly from theoretical studies and high-resolution numerical models. Submesoscale processes make an important contribution to the vertical fluxes of mass, buoyancy, and tracers in the upper ocean [31], and they are frequently dominant near ocean fronts, where high productivity has been reported [35–38]. It is thus not surprising that submesoscale variations near and within ocean fronts are among the important research topics recently targeted in physical, chemical, biological, and fisheries oceanography. A critical problem for such studies, however, has been the paucity of observational versus simulated data with which to identify submesoscale variations because high spatiotemporal variability confounds observing the entire structure of submesoscale variations.

A very basic question then arises as to whether submesoscale variations in the ocean can be detected by satellite-derived skin SST data; few reports have examined this problem. Here, we ask whether or to what extent skin SST data from Himawari 8 can capture submesoscale variations in the ocean. Although the 10-min temporal resolution of Himawari 8 sampling is much shorter than the timescale of submesoscale variations, the fact that the 2-km spatial resolution falls near the upper bound of the typical spatial range of these variations suggests that this satellite's spatial resolution may compromise such measurements. On the other hand, our understanding of the spatial scale of submesoscale variations that can be identified by satellite-derived SSTs is incomplete.

High-resolution skin SST data have been measured as a part of the Landsat program. The scene size of the latest Landsat 8 satellite is 185 km cross-track by 180 km along-track, and the spatial resolution

of the Thermal Infrared Sensor (TIRS), which is no more than 100 m, is comparable to the minimum scale of representative submesoscale ocean variations. Landsat 8 also has a 16-day repeat cycle with an equatorial crossing time. Landsat 8 data are therefore not appropriate for time series analyses at a frequency as often as daily or weekly, but they do an excellent job of providing comprehensive measurements of fine spatial structures (i.e., submesoscale variations) in an instantaneous snapshot. This capability is also much more powerful at the spatial scale of continental shelves (~$O(10^2)$ km)) than the capability of unmanned aerial vehicles, which have a higher spatial resolution but limited spatial coverage (e.g., 0.5-m spatial resolution and 100-m swath width [39]).

Note that Landsat-derived skin SSTs need to be manually corrected and validated for, inter alia, atmospheric corrections [40–42], partly because the TIRS of Landsat 8 has only two spectral channels—namely, Bands 10 and 11—and Band 11 may have a large calibration uncertainty [43]. For this reason, Landsat 8-derived skin SST data have seldom been used in oceanographic studies. Even when Landsat 8-derived skin SSTs have been used, the research purposes have been mainly to just describe the spatial variations of SST in coastal and estuarine waters, where other satellites have difficulty monitoring SSTs [44–49]. In sum, few studies have examined submesoscale variations based on horizontal gradients of Landsat 8-derived skin SSTs combined with in situ measurements. If these kinds of studies confirm that Landsat 8-derived skin SSTs can successfully capture submesoscale fronts, then Landsat 8-derived SSTs could be used to provide essential evidence of submesoscale processes. Moreover, these kinds of studies are imperative for determination of the grid size of ocean models that is best suited for the reproducibility of prominent submesoscale variations.

The goals of the present study were to propose a simple method to correct Landsat 8-derived skin SSTs by using the skin SSTs from Himawari 8 and to examine the properties of thermal frontal structures on the Pacific shelf south off the Hokkaido coast, Japan, during October 2016, in combination with data from in situ shipboard measurements (Figure 1). According to the satellite-based SST front analyses by Belkin and Cornillon [50], frontal structures are enhanced seasonally during August–October around the study area. Note that in the present study, we analyzed only a few snapshots of the skin SST images; thus, this study is regarded as the first step before statistical or dynamical analyses in the near future of submesoscale variations using multiple long-term images from Landsat 8.

This paper is structured as follows. Section 2 explains the data and data processing. In Section 3, snapshots of the skin SST on 13 October 2016 are validated by combining Landsat 8-derived skin SSTs with in situ ship measurements, and features of the thermal frontal structures along the Pacific shelf are examined. Section 4 provides a brief discussion, and Section 5 summarizes the new findings from this study.

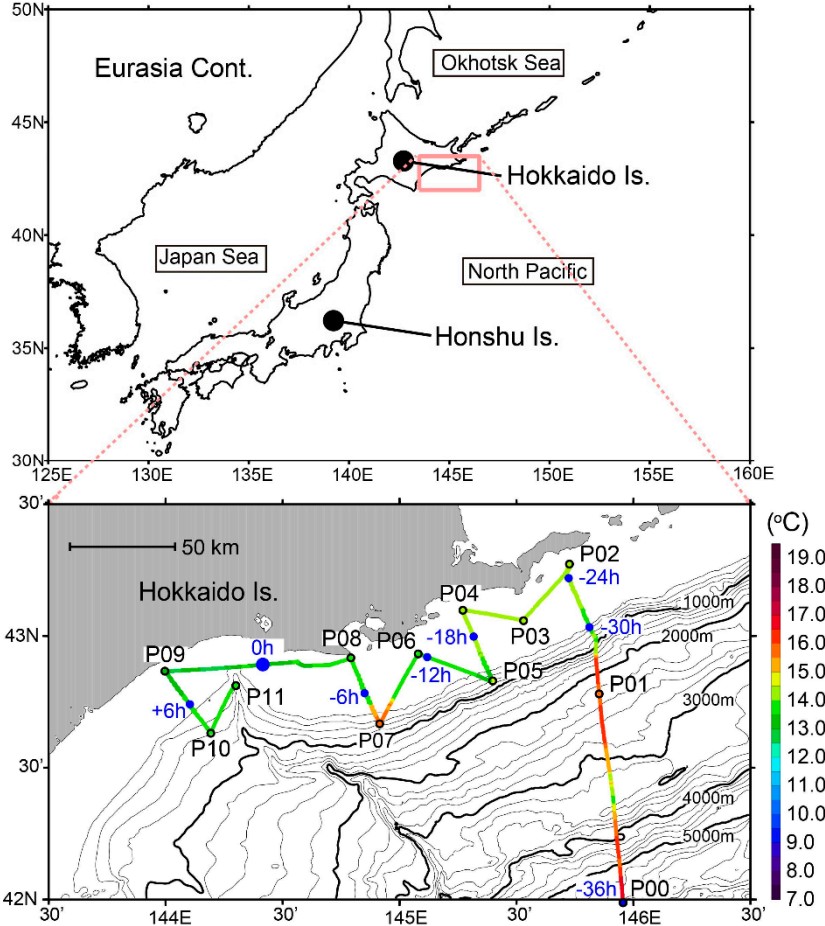

**Figure 1.** Study area around the Pacific shelf off the southeastern coast of Hokkaido. In lower panel, in situ SSTs along the ship track are indicated by the color scale at the right. Points P00–P11 are representative points along the ship track. Time of passage of the ship relative to an observation by Landsat 8 is indicated by blue closed circles with the time in hours.

## 2. Materials and Methods

### 2.1. Satellite-Derived SSTs

For correction and comparison of Landsat-derived skin SST data, we used high-quality skin SST data from Himawari 8, which is operated by the Japan Meteorological Agency (JMA) [8,51]. The skin SST data that we analyzed were referred to as "standard product (version 1.2)" and were collected from the P-Tree System of the Japan Aerospace Exploration Agency (JAXA). The spatial resolution was 2 km. We used 10-min, Level 2, retrieved skin SST data and 1-h, Level 3, composite skin SST data, the observation times of which were the closest to those of the Landsat 8-derived SSTs.

Landsat 8 data were downloaded from the archives of the United States Geological Survey (USGS) using the EarthExplorer user interface. The Landsat 8 carries onboard a TIRS with two spectral bands, Band 10 (10.6–11.19 μm) and Band 11 (11.5–12.51 μm) [52]. We collected Landsat Collection 1 Level-1 Data. The TIRS bands were originally acquired at 100-m resolution but resampled to 30-m resolution in producing the Level-1 Data. There was striping and banding noise in the TIRS [43]. Striping is a phenomenon that appears as columns of consistently lighter or darker pixels in a single band of radiometrically corrected data, and banding occurs across multiple contiguous columns. Striping and banding were the most problematic forms of noise and particularly confounded the estimation of horizontal gradients of the skin SST in this study. To mitigate the striping and banding problem, we applied a digital filter that was originally proposed by Crippen [53]. The digital value of a Level 1

pixel was converted to spectral radiance, and the spectral radiance was then converted to a brightness temperature, which is the effective temperature viewed by the satellite on the assumption of unit emissivity [43].

We corrected the brightness temperatures with respect to Bands 10 and 11 using skin SSTs from Himawari 8. The accuracy of Band 11 was reexamined in this study through comparison with that of Band 10. For each band, the brightness temperatures at the sea surface from Landsat 8 were spatially averaged for 2-km grid cells centered on the Himawari 8 grid points. A linear regression line was estimated by plotting the 2-km brightness temperatures from Landsat 8 (explanatory variable) versus the skin SSTs from Himawari 8 (dependent variable) (Figure 2). Cloud area was excluded from the regression analysis. Using the regression line, the brightness temperatures were converted to skin SSTs. In this study, we analyzed only a snapshot image for each band from Landsat 8 at 01:01 (UTC) on 13 October together with in situ measurements. At that time, there were relatively gentle winds near the sea surface; ~5 m s$^{-1}$ for shelf waters and ~7 m s$^{-1}$ for slope waters.

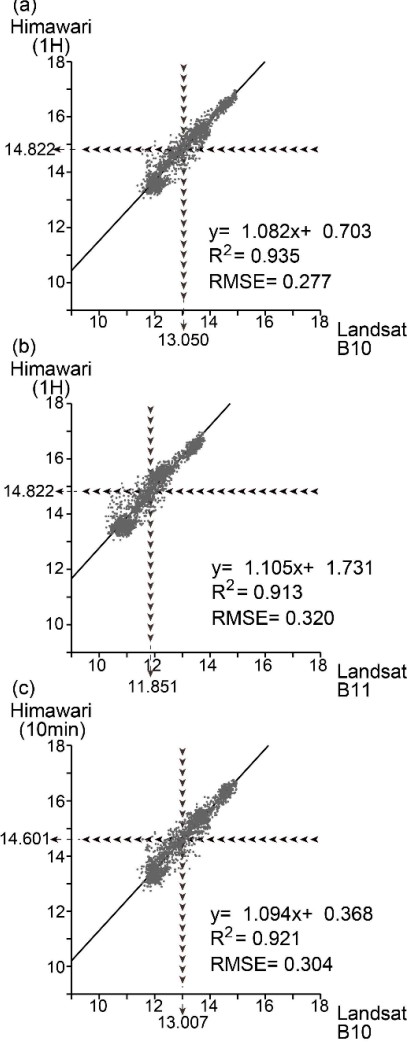

**Figure 2.** Scatter plots of skin SSTs based on different datasets. (**a**) Skin SSTs from Band 10 of Landsat 8 versus skin SSTs from 1-h composite of Himawari 8 images. (**b**) Same as (**a**), but for skin SSTs from Band 11 of Landsat 8. (**c**) Same as (**a**), but for skin SSTs from 10-min retrieve of Himawari 8 images. Individual data points are based on grid points of Himawari 8 images without clouds with a resolution of 2 km. Each panel shows an equation of the regression line, coefficient of determination, root mean square error (RMSE), and averages of the two variables.

*2.2. In Situ Measurements*

We performed ship surveys from 11 to 13 October 2016 around the Pacific shelf south of Hokkaido, Japan (Figure 1). Along the track of R/V Hokko-maru off the southeastern coast of Hokkaido in the North Pacific, temperatures near the sea surface and geolocations were continuously recorded by a digital thermometer and a global positioning system receiver, respectively, at intervals of 1 min from 13:00 (UTC) on 11 October to 13:00 (UTC) on 13 October 2016. Horizontal current velocities were simultaneously measured by a ship-mounted 150-kHz Acoustic Doppler Current Profiler (ADCP, RD Instruments). The survey started at the southeasternmost point P00 (Figure 1) on the continental slope and proceeded to the north. The ship then moved repeatedly to the southwest and northwest as it crossed thermal fronts on the shelf (Figure 1). Conductivity–temperature–depth (CTD) measurements were conducted along the across-shelf transects of P01–P02, P04–P05, P07–P08, and P09–P10 at intervals of about 2 km. A snapshot SST image of Landsat 8 was obtained when the vessel passed through the position that is indicated by the blue closed circle designated "0 h" in Figure 1. The thermometer temperatures were corrected using a regression line ($R^2$ = 0.997) that related the temperatures of water sampled by bucket from a few dozen centimeters below the sea surface to SST at the CTD stations. The ship speed changed from 0.1 to 16.1 knots during the period of analysis; the mean ship velocity was 7.2 knots. The spatial resolution of the ship-track SST data therefore changed from 3 to 497 m, with an average of 222 m, which is roughly comparable to the order of magnitude of the representative minimum scale of a submesoscale variation ($O$(0.1–10) km).

## 3. Results

*3.1. Comparison of SSTs among Datasets*

We created four linear regression models using two types of datasets from Landsat 8 (Bands 10 and 11) and Himawari 8 (1-h composite and 10-min retrieved skin SSTs) to find the best combination for correcting Landsat 8 data. Figure 2 shows three of the four regression models. The most robust regression model was obtained between the brightness temperatures from Band 10 and 1-h composite skin SSTs from Himawari 8 (Figure 2a). For that model, the coefficient of determination was the largest ($R^2$ = 0.94), and the RMSE was the smallest (0.28 °C). However, the regression line indicated that the brightness temperatures from Band 10 (mean SST = 13.05 °C) underestimated the skin SST from Himawari 8 (mean SST = 14.82 °C). This low-temperature bias relative to Himawari 8 was much more serious for brightness temperatures from Band 11 (mean SST = 11.85 °C) (Figure 2b). These results indicated that the brightness temperatures from the Landsat 8 needed to be corrected to obtain precise skin SSTs.

When we compared regression models based on different objective variables such as 1-h composite and 10-min retrieved data from Himawari 8 (Figure 2a,c), the 10-min retrieved data provided a less robust regression model. This difference implied that more accurate skin SSTs were estimated in the 1-h composite from Himawari 8 by merging six 10-min retrieved skin SST images to reduce cloud noise. The mean SST was 0.22 °C lower for the 10-min retrieved skin SSTs (14.60 °C) versus the 1-h composite skin SSTs (14.82 °C).

We corrected brightness temperatures from Band 10 using the regression line in Figure 2a, interpolated them onto the ship track (Figure 1), and compared the interpolated temperatures with in situ SSTs (Figure 3a). The corrected brightness temperatures from Landsat 8 are hereafter referred to as skin SSTs. The skin SSTs and in situ SSTs were very consistent near the time when the Landsat 8 image was obtained. There were large discrepancies, however, at times >6 h before the observation time of Landsat 8. These discrepancies suggested that the persistence of SSTs along the ship track was roughly <6 h for the obtained Landsat 8 image.

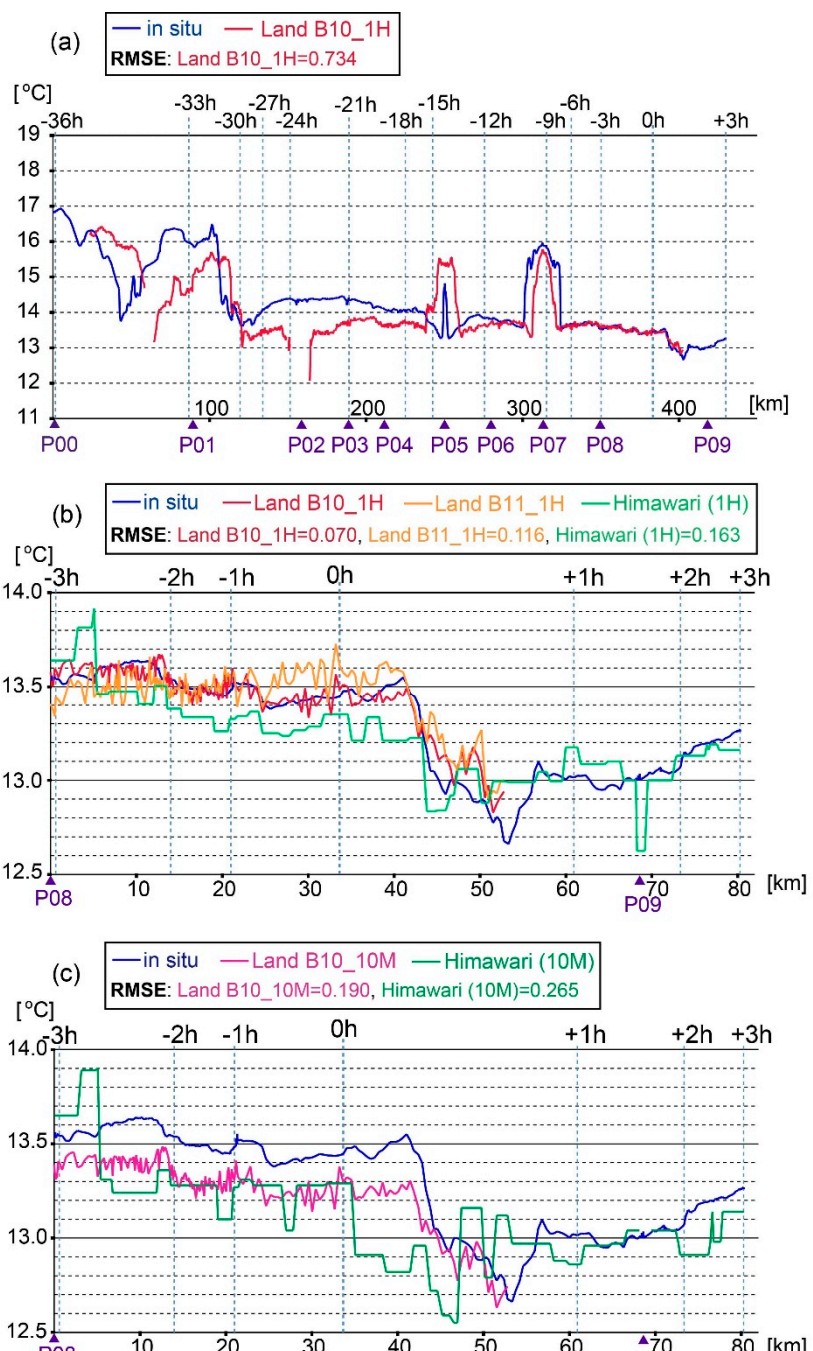

**Figure 3.** Comparison between in situ SSTs and several different satellite-derived skin SSTs along the ship track. Lower horizontal axis is based on the cruise distance of the ship. (**a**) From −36 to +3 h relative to an observation time of Landsat 8, in situ SSTs and skin SSTs from Band 10 that were corrected by 1-h composite skin SSTs estimated from Himawari 8 images are indicated by blue and red lines, respectively. (**b**) Same as (**a**), but for illustrated period of ±3 h and additional two satellite-derived skin SSTs: skin SSTs from 1-h composite of Himawari 8 images (green line) and skin SSTs from Band 11 that were corrected by the 1-h composite data (orange line). (**c**) Same as (**b**), but for 10-min retrieved skin SSTs from Himawari 8 images (green line) and skin SSTs from Band 10 that were corrected by the 10-min retrieval data (pink line). Skin SSTs from Band 11 are not depicted.

Some of the satellite-derived skin SSTs and in situ SSTs were compared along the ship track during the 3 h before and after the observation time of Landsat 8 (Figure 3b,c). Note that during the ±3-h

period, the research vessel approximately moved westward near the coast from P08 to P09 (Figures 1 and 3). Skin SSTs from Band 10 that were corrected by the 1-h composite skin SSTs from Himawari 8 (red line in Figure 3b) were the closest to in situ SSTs (blue line in Figure 3b). The RMSEs relative to the in situ SSTs were 0.070 °C for Band 10, 0.116 °C for Band 11, and 0.163 °C for the 1-h composite of Himawari 8 (Figure 3b). Skin SSTs from Band 11 included higher levels of small-scale noise than those from Band 10. The major reason why the skin SSTs from Landsat 8 were more precise than those from Himawari 8 is that the Landsat 8 measured skin SSTs with a resolution 20 times that of the Himawari 8 and was better able to measure skin SSTs around coastal waters.

The skin SSTs from Band 10 that were corrected by the 10-min retrieved skin SSTs (pink line in Figure 3c; RMSE = 0.190 °C) were less accurate than those that were corrected by the 1-h composite skin SSTs (red line in Figure 3b). The larger RMSE of 0.190 °C resulted mainly from the low-temperature bias of the corrected skin SSTs along the ship track (pink line in Figure 3c) compared with in situ SSTs (blue line in Figure 3c). Our correction method based on a regression line (Figure 2) eliminated SST biases between the Landsat 8 and Himawari 8 gridded datasets in the area where they overlapped without clouds. The low-temperature bias of the corrected skin SSTs (pink line in Figure 3c) could therefore be attributed to the mean SST difference (~0.22 °C) between the 1-h composite and 10-min retrieved data (e.g., Figure 2a,c), which were probably less accurate because of the inclusion of a higher level of cloud noise, as discussed above.

### 3.2. Structures of Thermal Fronts on the Shelf and Slope

Skin SSTs from Landsat 8 that are shown hereafter are the skin SSTs from Band 10 that were corrected by 1-h composite data from Himawari 8. Figure 4a,b show the maps of skin SSTs from the 1-h composites of Himawari 8 and Landsat 8, respectively. The overall spatial patterns of skin SSTs were consistent between the two datasets; bands of cold, warm, cold, and warm water that extended more-or-less parallel to the coastline were found sequentially in an onshore-offshore direction. However, as mentioned before, skin SSTs estimated from the Himawari 8 and Landsat 8 imagery differed near the coast. Skin SSTs from Landsat 8 captured small-scale variations there. In addition, the skin SSTs from Himawari 8 were more ambiguous around boundaries between cold and warm water than those from Landsat 8. Horizontal gradients of skin SSTs therefore differed markedly between the two datasets (Figure 4c,d).

In the skin SSTs from Himawari 8, there were approximately three SST fronts (defined as local maxima in horizontal SST gradients) along the coastline between the aforementioned bands of cold and warm water (Figure 4c). The SST front closest to the coast was on the continental shelf, and the rest were on the slope. Horizontal gradients along the three fronts were roughly the same order of magnitude.

In contrast, not all of the three fronts were apparent in skin SSTs from Landsat 8 (Figure 4d). Horizontal gradients near the SST fronts on the shelf (Figure 4d) were larger by one order of magnitude than those based on Himawari 8 (Figure 4c). The width of the SST fronts based on Landsat 8 was clearly narrower. These differences between Himawari 8 and Landsat 8 imagery indicated that the intensity and width of the SST fronts depended strongly on the horizontal resolution of the satellite data. In skin SSTs from Landsat 8, double frontal structures were identified partially on the shelf (Figure 5). The SST fronts were characterized by wavy structures along the frontal axis. The SST fronts were also ambiguous or discontinuous in some portions of the fronts. Moreover, the wavy structures associated with the frontal waves appeared to become smaller to the west.

It is noteworthy that the two fronts on the slope, which were apparent in skin SSTs from Himawari 8 (Figure 4c), were obscure in skin SSTs from Landsat 8 mainly because the signal-to-noise ratio was low in the Landsat 8 image (Figure 4d and lower right part of Figure 5), although there were certainly boundaries on the slope between cold and warm waters (Figure 4b). Weak fronts with gradients of $O(0.1)$ °C km$^{-1}$ were obscure in the Landsat 8 skin SSTs, whereas relatively strong fronts with gradients

of $O(1)$ °C km$^{-1}$, such as the fronts on the shelf, were apparent. Skin SSTs from Landsat 8 were thus suitable for detecting submesoscale fronts with relatively strong intensities.

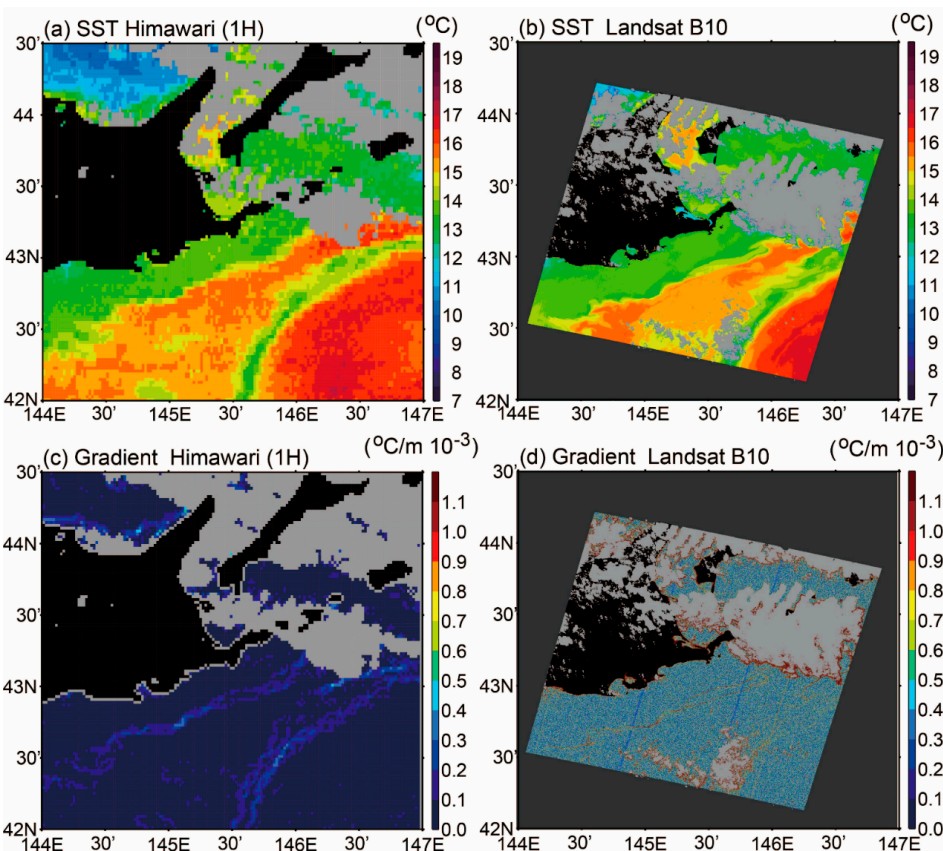

**Figure 4.** Maps of skin SSTs based on (**a**) 1-h composite of Himawari 8 images and (**b**) Band 10 of Landsat 8 that were corrected by the 1-h composite data. (**c**,**d**) Same as (**a**,**b**), but for horizontal gradients of skin SSTs (i.e., |∇SST|).

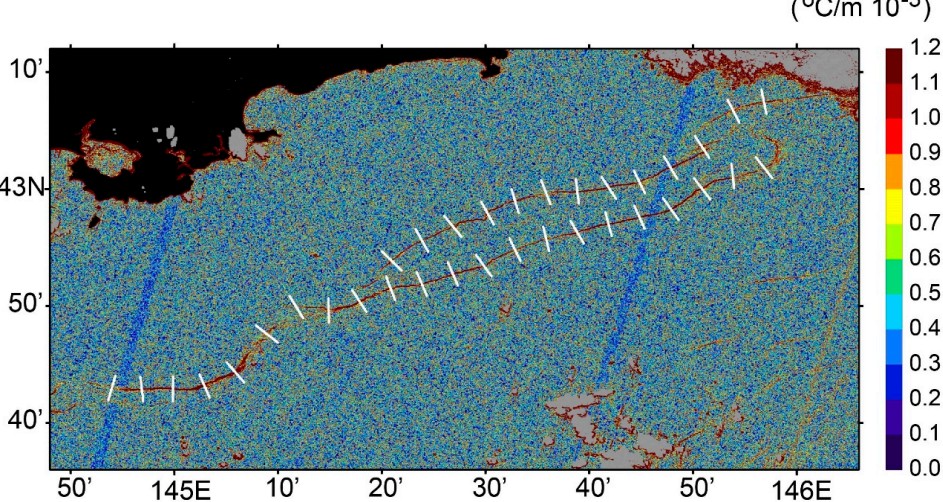

**Figure 5.** Same as Figure 4d, but for an enlarged region of the map. Thirty-five transects approximately normal to the SST fronts are denoted by white lines. Some statistical values are estimated along these transects in Figure 7e.

We compared SST fronts along the ship track on the shelf between the skin SSTs from Landsat 8 and the in situ SSTs (Figure 6). The research vessel crossed the SST fronts five times on the across-shelf transects of P01–P02, P04–P05, P05–P06, P06–P07, and P07–P08. The ship crossed transects P01–P02 and P07–P08 about 32.2 h and about 7.3 h, respectively, before the SST image was obtained by Landsat 8. Because the persistence of SSTs along the ship track was less than roughly 6 h for the Landsat 8 image that we obtained (Figure 3a), there were discrepancies between SST fronts measured in situ and estimated from Landsat 8 imagery. Those discrepancies were expected to increase with increased time between the two sets of observations.

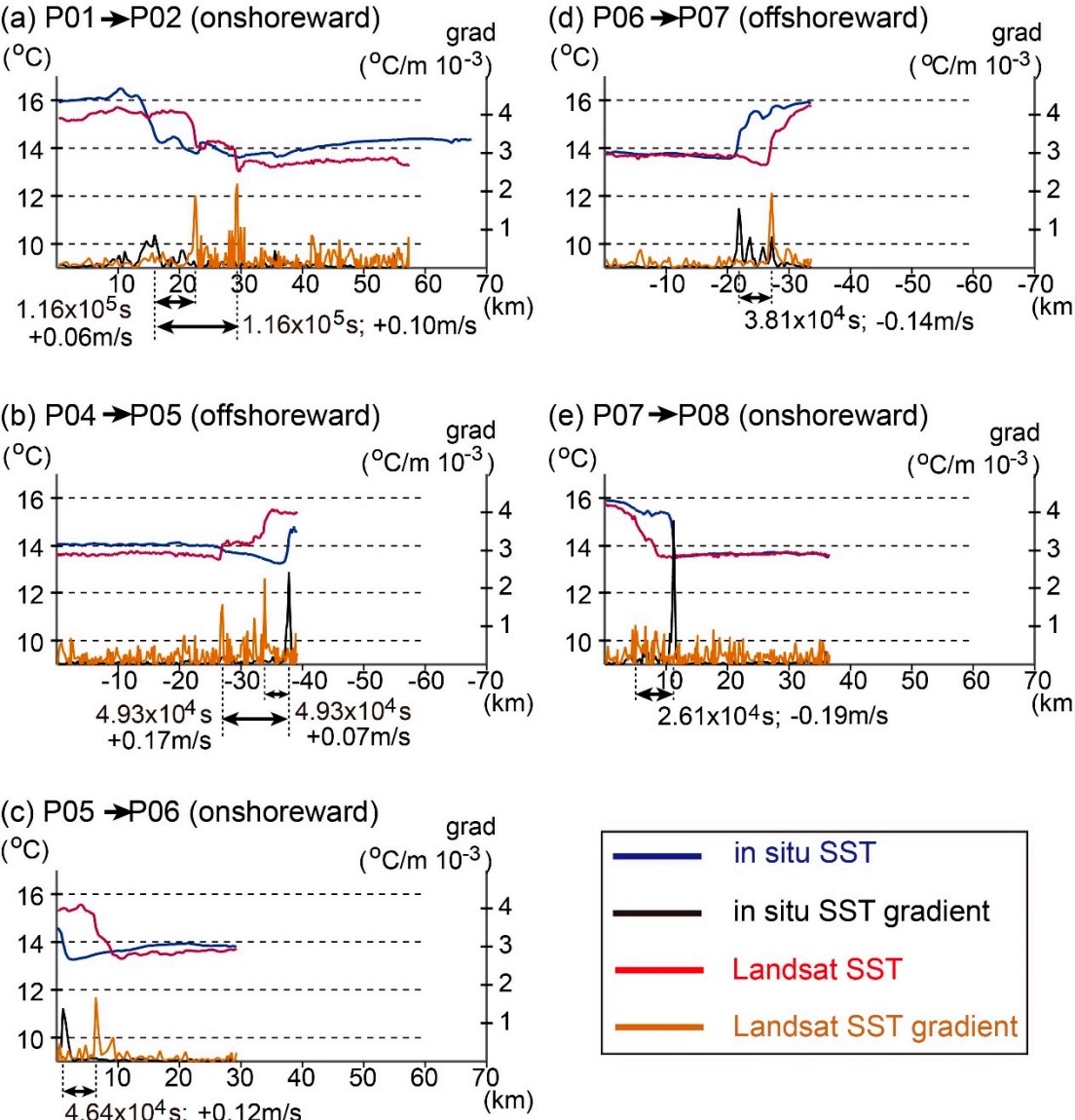

**Figure 6.** SSTs (**left vertical axis**) and their horizontal gradients (**right vertical axis**) along each transect of the ship; (**a**) P01–P02, (**b**) P04–P05, (**c**) P05–P06, (**d**) P06–P07, and (**e**) P07–P08. In situ SSTs and their gradients are denoted by blue and black lines, respectively. Landsat 8-derived skin SSTs and their gradients are indicated by red and orange lines, respectively. Positive (negative) cruise distance was defined when the ship moved in an onshore (offshore) direction, respectively. Double-headed arrows represent the difference in the positions of the front in Landsat 8 and in situ SST estimates, and the accompanying numbers indicate the temporal difference between the two observations and the apparent onshore/offshore speed of movement of the SST front.

Along transect P01–P02 (Figure 6a), the SSTs and temperature gradients from the two datasets were not in good agreement; the in situ SSTs exhibited single fronts, whereas the Landsat 8-derived skin SSTs were characterized by double fronts with two step-like jumps. These discrepancies were attributable to the time difference of 32.2 h between the two observations. The implication is that it is better to exclude SST fronts along transect P01–P02 when comparing the two datasets.

The temporal difference was much shorter (13.7 h) for the SST fronts along transect P04–P05 (Figure 6b). As was the case on transect P01–P02, in situ SSTs exhibited a single front, and Landsat 8-derived skin SSTs had double fronts. The local maximum of the SST gradient at the front was comparable between the front of the in situ SSTs and the offshore portion of the Landsat 8 double fronts. The speeds of the fronts toward the shore were tentatively estimated to be 0.07 and 0.17 m s$^{-1}$ on the basis of the differences of the observed frontal positions and the time interval between the two sets of SSTs. Along transects P05–P06 (Figure 6c) and P06–P07 (Figure 6d), the single fronts were identifiable from the two datasets, and the maximum temperature gradients were comparable between them. Along transect P05–P06, the SST fronts were 5344 m apart and represented onshore movement at 0.12 m s$^{-1}$, and along transect P06–P07 they were 5300 m apart and represented offshore movement at 0.14 m s$^{-1}$. Along transect P07–P08, the Landsat 8 SST front was clearly broader than the in situ SST front, despite the fact that the temporal difference was only 7.3 h between the two observations (Figure 6e); the maximum gradient at the front based on Landsat 8-derived skin SSTs was about 20% of the corresponding gradient based on in situ SSTs. The distance between the SST fronts in the two datasets was 5090 m, from which the speed of movement offshore was estimated to be 0.19 m s$^{-1}$.

We used the distribution of the horizontal gradients of in situ SST along each across-shelf transect to estimate the widths of the thermal fronts (Figure 7a–d). The width of a front was defined by the e-folding scale of the height/difference between the local maximum of the SST gradients and the baseline. The widths of the SST fronts were estimated to be 539–1050 m. The frontal widths estimated from in situ SSTs were therefore clearly narrower than the horizontal resolution of skin SSTs estimated from Himawari 8 images and fell within but near the lower bound of the horizontal dimensions of submesoscale fronts ($O$(0.1–10) km).

To describe oceanographic relationships between physical conditions at the sea surface and in the subsurface near SST fronts, we examined vertical sections of CTD-based temperatures and ADCP-based current velocities along the P07–P08 transect (Figure 8). A layer of vertically homogeneous temperatures at depths of 30–50 m below the sea surface was associated with development of a surface mixed layer due to vertical convection that was driven by sea surface cooling in autumn. The thermal front that was detected from in situ SSTs (Figures 6e and 7d) was not limited to the sea surface. The thermal front also corresponded to a density front (not shown). The dominant longshore currents flowed in a west-southwestward direction and became more intense near the sea surface around the thermal front. This structure was associated with a typical baroclinic jet on the shelf. The maximum speed of the baroclinic jet exceeded 0.7 m s$^{-1}$ (Figure 8b). SST fronts detected in situ and by Landsat 8 on the shelf (Figures 6e and 7d) were therefore tightly linked with the subsurface structure related to the baroclinic jet.

To estimate the widths of SST fronts from the Landsat 8 image, we analyzed data from 35 transects perpendicular to the fronts at almost evenly spaced intervals (Figure 5). Horizontal gradients of skin SSTs were plotted as a function of distance from the fronts along each transect (Figure 7e). Before describing the widths of the SST fronts, we should note that the mean baseline value (the lowest of the three green, dashed lines in Figure 7e) was estimated to be 0.5 °C km$^{-1}$. This value corresponded to the typical noise level of the horizontal gradients of Landsat 8-derived SSTs in an area without remarkable fronts. The noise level of Landsat 8 was at least five times the corresponding noise level of Himawari 8, which was less than 0.1 °C km$^{-1}$ in an area without fronts (Figure 4c).

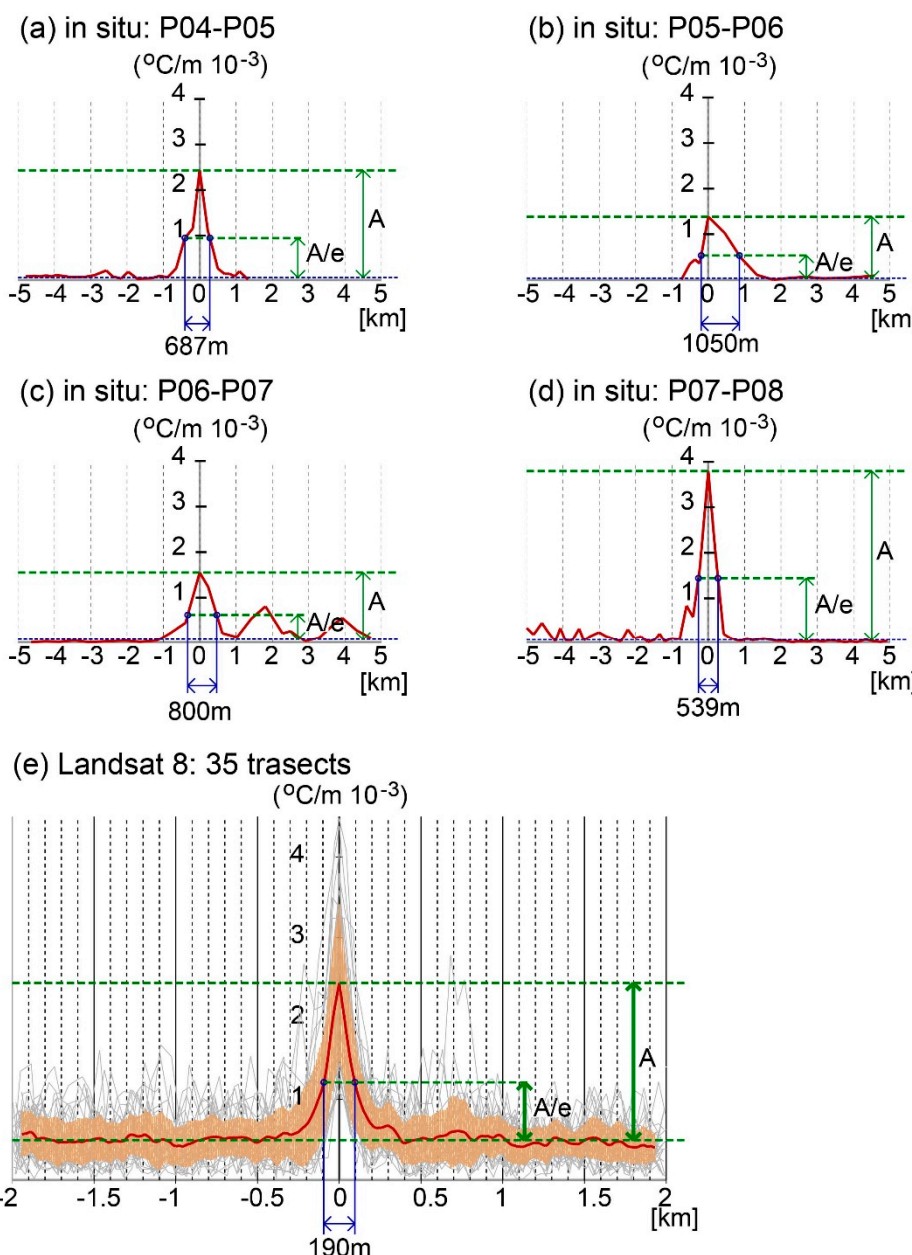

**Figure 7.** Plots showing the horizontal gradient of in situ SST (red line) as a function of distance from the front for transects (**a**) P04–P05, (**b**) P05–P06, (**c**) P06–P07, and (**d**) P07–P08. The height of the peak in panels (**a**)–(**d**) is the difference between the maximum and the baseline of this line, and the quotient of this height divided by e (the base of the natural logarithms) is the e-folding scale, from which the width of the front (blue double-headed arrow) was estimated. (**e**) Horizontal gradients of skin SSTs from Landsat 8 across 35 transects (see Figure 5). Gray, red, and orange lines indicate horizontal gradients for individual transects, mean, and standard deviation of the 35 transects, respectively.

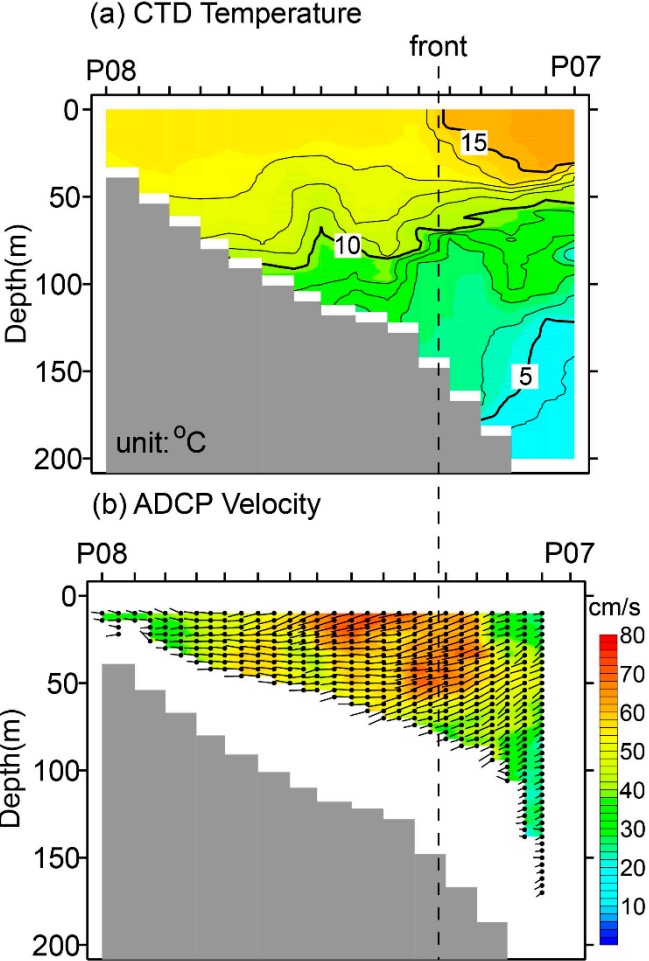

**Figure 8.** Vertical sections of (**a**) CTD-based temperatures (at ~2 km intervals) and (**b**) ADCP velocity (arrows) and speed (colors) along transect P07–P08. Vertical dashed line represents the approximate position of the thermal front.

From the mean distribution of the gradients (red line in Figure 7e), the width of the SST front based on the e-folding scale was estimated to be 190 m. When we estimated the width of the SST front from the distribution of SST gradients along each transect, the minimum, maximum, mean, and standard deviation of the 35 transects were 114, 461, 215, and 73 m, respectively. The widths of the SST fronts estimated from Landsat 8 imagery were thus narrower than those estimated from in situ SSTs. This discrepancy was probably attributable to the difference of the depths of the SST measurements, which were about a 10-μm skin layer in the case of the satellite and a few dozen centimeters below the sea surface in the case of the in situ survey, as discussed in the next section. In brief, in situ SSTs and skin SSTs from Landsat 8 imagery indicated that the width of the SST front on the shelf was 100–1000 m. These results implied that an ocean model with a grid size ≤100–200 m is essential for simulating the realistic frontal structure on the Pacific shelf.

## 4. Discussion

We first discuss the possible causes of the discrepancy between the widths of the SST fronts estimated from Landsat 8 imagery and the in situ SSTs (Figure 7). The discrepancy was probably due to the differences between skin and bulk SSTs, in addition to the differences of observation times between in situ and satellite SSTs for each front. In general, the in situ SST measured at a depth of about 1 m or deeper has been called the "bulk" SST [54]. Differences between skin and bulk SSTs have frequently been observed over the world ocean. Skin-bulk SST differences depend strongly on net

heat fluxes and wind speeds at the sea surface [55–59]. Many previous studies have indicated that the skin-bulk SST difference vanishes under strong-wind conditions (>7–10 m s$^{-1}$), during which the ocean skin layer is disrupted by wave breaking. In contrast, under low-wind conditions (typically, <4–5 m s$^{-1}$), a skin layer is established/maintained, and a skin-bulk SST difference therefore occurs. In our case, when the Landsat 8 snapshot image was obtained, there were moderate winds near the sea surface; ~5 m s$^{-1}$ for shelf waters and ~7 m s$^{-1}$ for slope waters, as explained before. The fact that the wind speeds at that time fell in the transition regime between low and high wind conditions suggests that skin layers were barely maintained on the Pacific shelf at the time of the Landsat 8 observations. In addition, the observation time was about 4.5 h after sunrise. At that time of day, solar radiation fluxes through a clear sky to the sea surface increase gradually toward noon and may have contributed to the maintenance of the skin layer by heating.

We next discuss a practical strategy that might be used to enhance understanding of the properties and dynamics of submesoscale fronts on the Pacific shelf by using satellite-derived SSTs. Particularly important points are that the thermal fronts observed on the Pacific shelf were spatially inhomogeneous, and they were transported along the shelf by a baroclinic jet. Regarding the inhomogeneity, the strong SST fronts ($O(1)$ °C km$^{-1}$) on the shelf showed single or double frontal structures (Figure 5). There were wavy structures along the frontal axis and ambiguous or discontinuous structures in some portions of the fronts. There were large deviations among the frontal widths, even for the 35 transects along which distinguishable fronts could be identified (i.e., minimum 114 m; maximum 461 m; mean of 215 m; and standard deviation 73 m). Moreover, the fact that the widths of the fronts fell within the spatial scale typical of submesoscale variations (i.e., $O(0.1–10)$ km) suggests that the fronts changed on a timescale of days. This timescale suggests why it is difficult to continuously observe spatiotemporal variations of the intensity and position of such fronts using only available satellite images unless geostationary satellites can measure SSTs with a resolution <100 m.

Moreover, the speeds of onshore/offshore movement of SST fronts along each transect were estimated to be 0.07–0.19 m s$^{-1}$ based on the spatiotemporal differences of the across-shelf positions of the fronts between in situ observations and Landsat 8 imagery (Figure 6). These speeds were small compared to the maximum speed of the strong longshore currents that were related to a baroclinic jet around the SST fronts (Figure 8); those longshore current speeds exceeded 0.7 m s$^{-1}$ near the fronts and were 3.5–10 times the speed of movement of the fronts across the shelf. If the fronts detected by in situ SST measurements on transects P05–P06, P06–P07, and P07–P08 were transported by a longshore current at a speed of 0.7 m s$^{-1}$, they would have moved downstream by 32, 27, and 18 km, respectively, before the time when the Landsat 8 image was obtained. Moreover, along transect P07–P08, the SST front detected in the Landsat 8 image was much broader than the in situ SST front (Figure 6e). However, similarly broad fronts were not identified from in situ SSTs upstream/east of transect P07–P08 (Figure 7a–d). Accordingly, the much broader fronts detected in the Landsat 8 image suggest that the SST front was greatly diffused as it was transported by the longshore current during the 7.3 h that separated the two observation times. The biggest challenge in studying submesoscale frontal processes on the Pacific shelf is therefore the ability to capture the fronts that are transported by background jets while their intensity and width change spatiotemporally. Understanding the spatiotemporal variability of the thermal fronts embedded in baroclinic jets will therefore require repeated measurements by a ship or autonomous underwater vehicle on transects that cross the fronts [60]. Such measurements will be limited spatially but would be informative when combined with Landsat 8 and other satellite measurements targeting submesoscale variations (e.g., Isoguchi et al. [61]) and numerical models with a grid size of no more than ≤100–200 m.

## 5. Conclusions

We proposed a very simple method to correct skin SSTs from Landsat 8 imagery by comparisons with skin SSTs from Himawari 8, and we described the horizontal structure of SST fronts on the Pacific shelf off the Hokkaido coast of Japan, together with in situ SSTs measured along the ship track.

There have been some studies using a series of Landsat data to examine SST conditions in coastal waters, but to our knowledge, this study is the first to compare snapshots of submesoscale frontal structures in shelf waters with the results of in situ surveys. Skin SSTs estimated from Himawari 8 images greatly facilitated understanding the overall spatial pattern of the SSTs, except in the vicinity of the coast, whereas frontal structures with a large thermal gradient of $O(1)$ °C km$^{-1}$ were quantified less accurately than was possible with skin SSTs estimated from Landsat 8 imagery. It should be emphasized that the widths of SST fronts on the shelf that were estimated by in situ measurements and Landsat 8 imagery fell in the range 100–1000 m. Realistic simulation of the thermal fronts on the shelf will therefore require an ocean model with a grid size ≤100–200 m. The horizontal resolutions of operational ocean forecast systems for coastal waters around Japan are currently no less than about 2–3 km [62,63]. It is therefore likely that further higher-resolution modeling and forecasting can improve the reproducibility of ocean processes on the shelf. It should be noted again that this study dealt with only one snapshot of skin SST from a Landsat 8 image. Our proposed methodology can be used to process many skin SST images and to realistically compare those results with estimates based on high-resolution ocean modeling to better understand submesoscale processes and improve ocean models.

**Author Contributions:** Conceptualization, H.K.; formal analysis, Y.T.; data curation, H.K. and Y.T.; writing—original draft preparation, H.K.; writing—review and editing, H.K.; visualization, Y.T.; funding acquisition, H.K. All authors have read and agreed to the published version of the manuscript.

**Funding:** This work was supported by Management Expenses Grants from the Japan Fisheries Research and Education Agency and by the Ministry of Education, Science, and Culture (KAKEN grant 17H00775 and 19K06216).

**Acknowledgments:** We are deeply grateful to Satoshi Nakada, the National Institute for Environmental Studies, Japan, for an initial idea combining skin SST data from Landsat 8 and Himawari 8. We express our gratitude to the captain, officers, and crew of R/V Hokko-maru, as well as to researchers who supported the in situ measurements. We also extend a special thanks to editor, the associate editor, and two reviewers for their encouragement and their many constructive comments.

**Conflicts of Interest:** The authors declare no conflict of interest.

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
