# Peer review of "High-Resolution Sea Surface Temperatures Derived from Landsat 8: A Study of Submesoscale Frontal Structures on the Pacific Shelf off the Hokkaido Coast, Japan"

_remotesensing, doi:10.3390/rs12203326_

Round 1

Reviewer 1 Report

Dear Authors ...

Dear Authors ...

Introduction: Incorporating the history of SST sensors and new sensors as a background to the state of the art is fine but in a few lines. I prefer to see the antecedents of previous studies in the study area in order to give importance to the analysis presented. Precisely that they incorporate the studies of the analysis of mesoscale and submescale phenomena in addition to the ecological, biological and physical implications in the region of these types of structures. Expand the justification for using the images raised as well as their application in the development of oceanography and resource environment relationships.

In addition to the studies of the oceanographic dynamics of the zone adjacent to the study area to be able to explain these changes.

In this way, this study will have a greater impact and not only focus on correction methods.

Methodology. Regarding the methodology, in addition to the shortness of the study series, I would like to see an analysis of the frequency and duration of the submescale structures, I do not agree with such short studies since every oceanic or coastal zone changes constantly and with this analysis I do not see a continuity or application for the different relationships that can be applied with the oceanographic study with the images. I believe that with the results obtained from the study, the authors could have made the analysis longer and could have hard data which are applicable to other disciplines.

Therefore it is difficult for me as a reader to focus on reading an article that from my experience I cannot apply in any of the study of time series and the relationship of oceanographic dynamics through satellite images I do not see the application I think there are more technical magazines where this manuscript can be.

I hope you can make a greater effort and try to apply your analysis with the results that have or at least characterize all the structures, frequency and duration, even without having the relationship with the data in situ. That is the advantage of the analysis of satellite images.

Author Response

Could you please the attachment?

Reviewer 2 Report

See my comments in the attached file.

Author Response

Could you please see the attachment?

Round 2

Reviewer 1 Report

Dear Authors ...

I believe the modifications added to the manuscript are sufficient, and as you comment the application of this work is in the future, therefore it is important to disseminate it widely, it will be of great importance specifically to researchers who study environment-resource relationships and oceanographic dynamics. Likewise, I recommend to continue with the analyzes with longer time series and to be able to explain more clearly changes in the coastal areas with this time of application.

Author Response

Thank you very much for your comments.